# Potential Role of Global Longitudinal Strain in Cardiac and Oncological Patients Undergoing Cardio-Oncology Rehabilitation (CORE)

**Gianluigi Cuomo** [1,†]**, Francesca Paola Iannone** [2,†]**, Anna Di Lorenzo** [1]**, Crescenzo Testa** [3]**, Michele Ciccarelli** [4]**, Elio Venturini** [5]**, Arturo Cesaro** [6,7]**, Mario Pacileo** [8]**, Ercole Tagliamonte** [8]**, Antonello D'Andrea** [8,9]**, Carmine Vecchione** [4,10]**, Carlo Vigorito** [1] **and Francesco Giallauria** [1,*]

1. Department of Translational Medical Sciences, University of Naples Federico II, Naples, Via S. Pansini 5, 80131 Naples, Italy
2. Department of Clinical Medicine and Surgery, University of Naples Federico II, Naples, Via S. Pansini 5, 80131 Naples, Italy
3. Geriatric Clinic Unit, Geriatric-Rehabilitation Department, University Hospital, 43126 Parma, Italy
4. Department of Medicine, Surgery and Dentistry, Schola Medica Salernitana, University of Salerno, 84081 Baronissi, Italy
5. Cardiac Rehabilitation Unit, Azienda USL Toscana Nord-Ovest, Cecina Civil Hospital, 57023 Cecina, Italy
6. Department of Translational Medical Sciences, University of Campania "Luigi Vanvitelli", 80131 Naples, Italy
7. Division of Cardiology, A.O.R.N. "Sant'Anna e San Sebastiano", 81100 Caserta, Italy
8. Unit of Cardiology and Intensive Coronary Care, "Umberto I" Hospital, 84014 Nocera Inferiore, Italy
9. Unit of Cardiology, Department of Traslational Medical Sciences, University of Campania "Luigi Vanvitelli", Monaldi Hospital, 80131 Naples, Italy
10. Vascular Pathophysiology Unit, IRCCS Neuromed, 86077 Pozzilli, Italy
* Correspondence: francesco.giallauria@unina.it
† These authors contributed equally to this work.

**Abstract:** Although shown to be effective in improving survival and quality of life in patients with cancer, some treatments are well-known causes of cardiotoxicity, such as anthracyclines, monoclonal antibodies against human epidermal growth factor receptor 2 (HER2) and radiotherapy. To prevent cardiovascular disease (CVD) in patients living with cancer, cardiologists and oncologists promoted the development of cardio-oncology, an interdisciplinary field which aims to further improving life expectancy in these patients. Cardio-oncology rehabilitation (CORE), through correction of risk factors, prescription of drug therapies and structured exercise programs, tries to improve symptoms, quality of life, cardiorespiratory fitness (CRF) and survival in patients with cancer. Different imaging modalities can be used to evaluate the real effectiveness of exercise training on cardiac function. Among these, the global longitudinal strain (GLS) has recently aroused interest, thanks to its high sensitivity and specificity for cardiac dysfunction detection due to advanced ultrasound programs. This review summarizes the evidence on the usefulness of GLS in patients with cancer undergoing cardiac rehabilitation programs.

**Keywords:** cardiac rehabilitation; cancer; cardiotoxicity; cardio-oncology rehabilitation; global longitudinal strain; exercise training

## 1. Introduction

In many countries, cancer is the leading cause of death in people before the age of 70, with an estimated 19.3 million new cases and 10 million deaths [1]: in particular, female breast cancer was estimated to be the most diagnosed cancer (2.3 million new cases in 2020, 11.7% of total) [1].

Advances in care have increased the number of patients living with cancer worldwide [2,3], but these patients frequently develop cardiovascular disease (CVD) [4,5], including ischemic heart disease, stroke or heart failure (HF).

Patients undergoing some cancer therapies, such as anthracyclines, monoclonal antibodies against human epidermal growth factor receptor 2 (HER2) and chest radiation, are at greatest risk [6–8]. For example, women with breast cancer showed higher absolute risk for CVD death compared to the general population [9].

In the United States, patients living with cancer showed an increased prevalence of myocardial infarction (8.8% vs. 3.2%, $p < 0.001$), which is associated with increased financial worry and financial burden of medical bills ($p < 0.001$) [10].

Therefore, the need for effective management of both cancer and CVD for these patients has led to the development of cardio-oncology, an interdisciplinary specialty made up of experts who aim to provide a consistent, continuous and coordinated care to further increase life expectancy in patients living with cancer [11]. The growing interest in cardio-oncology has also recently led to the creation of guidelines by the European Society of Cardiology (ESC) [12].

Cancer-therapy-related cardiovascular toxicity (CTR-CVT) refers to the various CV manifestations, which may arise during or after use of cardiotoxic drugs or chest radiation, including cardiac dysfunction (i.e., HF or myocarditis), hypertension, vascular toxicity (coronary and peripheral artery disease, venous thrombosis), cardiac arrhythmias and pericardial and valvular heart disease [13]. In recent years, different classifications have been used to define the severity of cancer-therapy-related cardiac dysfunction (CTRCD). The most recent consensus statement of the International Cardio-Oncology Society (IC-OS) in 2021 divided CTRCD into a symptomatic and an asymptomatic form: the latter implies an impairment in left ventricular ejection fraction (LVEF) or in global longitudinal strain (GLS) deformation [13].

The severity and likelihood of clinical manifestation are also in part due to the presence of well-known CV risk factors, which may be pre-existing or consequent to anticancer therapy, particularly smoking [14] (which is also a risk factor for the development of cancer [15]), obesity [16], arterial hypertension [17,18], diabetes mellitus [19,20] and dyslipidemia [21,22]. A retrospective cohort study in 36,232 adult-onset ($\geq$40 years old) patients living with cancer showed that these patients had a significantly higher presence of hypertension, diabetes and dyslipidemia compared with the control population, and a greater CV risk was found in survivors of multiple myeloma (IRR, 1.70; $p < 0.01$), lung cancer (IRR, 1.58; $p < 0.01$), non-Hodgkin lymphoma (IRR, 1.41; $p < 0.01$), ovarian cancer (IRR, 1.41; $p < 0.02$), kidney cancer (IRR, 1.24; $p < 0.03$) and breast cancer (IRR, 1.13; $p < 0.01$) [5].

Therefore, the purpose of cardio-oncology is to identify both patients who experienced CTR-CVT and those at higher risk to develop CVD before cancer treatment to plan the most appropriate strategy [11].

Cardio-oncology uses imaging methods that are more suitable for detecting early CTRCD, including echocardiography and GLS [23,24]. In addition to initiating drug therapy, patients at higher risk are advised to undergo cardio-oncology rehabilitation (CORE) programs to prevent functional decline associated with cancer and its treatment [25]. The present review focused on the role of exercise-based and CORE programs on the principles of the GLS method and its current role and possible future utilization in patients with cancer.

## 2. Cardio-Oncology Rehabilitation (CORE)

Rehabilitation has long been used in patients with cancer to recover some specific lost functions and to provide psychological support [26,27].

Aiming to reduce cancer recurrence and complications, the American Cancer Society (ACS) published guidelines to provide best evidence on nutrition, weight management and physical activity for patients living with cancer [28].

The finding of increased CV risk in cancer survivors led to the adoption of an integrated rehabilitation approach in these patients based on the cardiac rehabilitation model, which contributed to a reduction in mortality and hospitalizations in patients with coronary heart disease and HF through both exercise training (ET) and CV risk factor management [29,30]. Moreover, in these patients, ET proved benefits on autonomic function [31–34], left atrial and left ventricle (LV) remodeling [35–38] and myocardial perfusion [39–41]. In addition, cardiac rehabilitation programs are cost-effective in patients with CVDs [42,43], showing significant improvement in quality-adjusted life year (QALY) (SMD: $-1.78$; 95% CI: $-2.69$, $-0.87$) and cost/QALY (SMD: $-0.31$; 95% CI: $-0.53$, $-0.09$) compared to the usual care for patients with coronary artery disease [42].

The new CORE concept aroused great attention, leading to the recommendation (class IIa) for cardiac rehabilitation in patients living with cancer with high CV risk by the most recent ESC guidelines on cardio-oncology [12]. However, the lack of robust evidence on the impact of CORE programs has been highlighted [12].

Back in 2019, the American Heart Association (AHA) published a scientific statement, proposing criteria for the selection of patients with cancer who should be candidates for cardiac rehabilitation, highlighting its benefits [44].

Although attention on cardiac toxicity has focused predominantly on CTRCD, the direct and indirect side effects of anti-cancer therapies are extended to the entire cardio-skeletal–muscle system. In fact, patients living with cancer may be affected by impairment in cardiorespiratory fitness (CRF) (i.e., the capacity of the circulatory and respiratory system to supply oxygen to skeletal muscles during physical activity), which is caused by the combination of age, physical inactivity, pre-existing comorbidities and cancer treatment side effects [45–48]. In this respect, it is particularly interesting that peak oxygen consumption ($VO_2$ peak) is a strong independent predictor of survival in some patients with cancer [49].

Current evidence shows that ET in patients with cancer is safe and effective in attenuating the increase in CV risk induced by cancer therapy [50]. Not only did exercise prove to counteract CV risk factors and consequently the risk of developing CVD calculated by the Framingham Risk Score [51], it also showed significant improvement in CRF ($+2.13$ mL $O_2 \times kg^{-1} \times min^{-1}$; 95% CI, 1.58 to 2.67; $I^2$, 20.6; $p < 0.001$) compared to usual care [52].

Different types of exercise may be used, such as high-intensity interval training (HIIT) and moderate-intensity continuous training (MICT), with similar improvements in $VO_2$ peak and peak oxygen pulse [53].

Although CRF is not a traditional parameter of CV risk, it was found to be a significant end point in patients living with cancer because its reduction was associated with a higher incidence of CTR-CVT, greater symptom burden and increased all-cause mortality [54–56].

Moreover, several other ET benefits have been found in patients with cancer. In breast cancer survivors, MICT showed to reduce markers of inflammation, such as high-mobility group box 1 protein (HMGB1) [57], and to improve endothelial function [58] and heart rate recovery (HRR) [59], which is the fall in heart rate during the first minute after exercise and is a known predictor of mortality.

Furthermore, exercise interventions in patients with breast cancer showed positive cost-effectiveness results, proving to be excellent public health investments [60].

CORE is a multidisciplinary intervention, which integrates exercise training prescription with other helpful intervention, such as psychosocial support, CV risk factors management and lifestyle counseling.

Emotional suffering in patients with cancer and their families is not only caused by fear for life expectancy, but also by the impact of cancer on daily routine (work, childcare, relationships, etc.). It has been estimated that about 15% and 10% of patients experience, respectively, anxiety and depression during treatments [61,62], with consequent impairment in health-related quality of life [63,64]. The cardiac rehabilitation model has been proven to improve quality of life and depression scores, in addition to CRF, in breast cancer survivors [65]. Moreover, techniques such as mindfulness showed reduction in anxiety and depression [66] and should be encouraged in CORE programs.

Weight loss is a goal to be achieved in all patients, since obesity, a well-known CV risk factor [67], is associated with poorer survival in patients living with cancer [68].

Furthermore, cardiac rehabilitation programs encourage intake of Mediterranean or vegetarian diets, which possess antioxidant and anti-inflammatory effects [69–71].

Figure 1 illustrates mechanisms of CORE in counteracting side effects of cancer and its therapy.

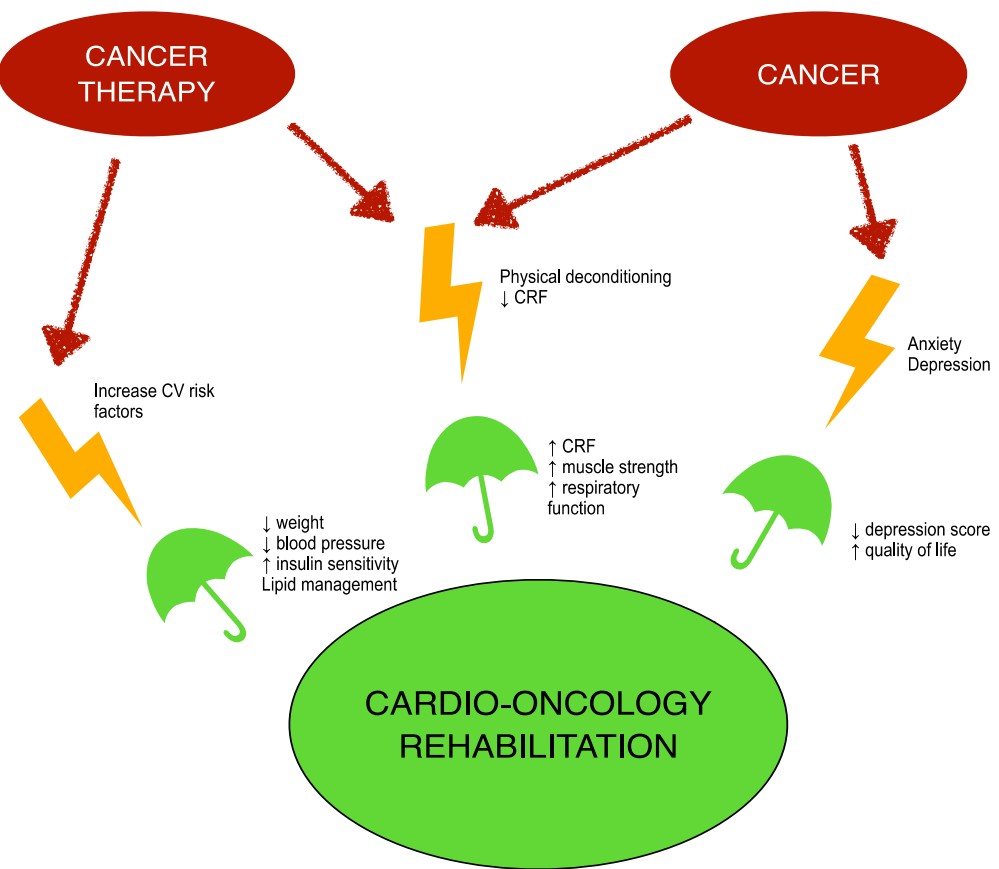

**Figure 1.** Mechanisms of CORE in counteracting side effects of cancer and its therapy. CRF: cardiorespiratory fitness; CV: cardiovascular.

Although various training protocols have been proposed [25,72], there is still a need to ameliorate patients' referrals and standardize specific rehabilitation programs [44].

It remains unclear which patients should undergo a cardiac rehabilitation program and whether larger benefits will be gained by starting this program before or after cardiotoxic therapy. Unfortunately, exercise programs are not feasible for all patients with cancer, since many barriers to CR exist in some populations, such as elderly and frail patients.

### 3. Global Longitudinal Strain (GLS), a New Technology for Detection of Systolic Disfunction

Since its outset, one of the main aims of echocardiography was the evaluation of cardiac function, using estimation of LV ejection fraction (LVEF), qualitative kinesis modifications and measurements of diastolic filling.

However, these measurements only partially describe the complex LV myocardium deformation during systole, which is characterized by longitudinal and circumferential shortening and radial thickening on the long axis, while the apex rotates in an anticlockwise rotation, and the base rotates in the reverse direction.

The procedure 2D speckle tracking echocardiography (STE) is an ultrasound technique that allows assessment of longitudinal, circumferential, and radial strain parameters of LV contractile function. The myocardial deformation is quantified as the percentage change in the length of a myocardial segment compared to its length at rest, using movement of bright speckle generated by scatter of the ultrasound beam [73].

Particularly, the longitudinal component of myocardial function, quantified by global longitudinal strain (GLS), has proved to have high sensitivity and specificity for detection of LV systolic dysfunction [74].

GLS is expressed by a percentage number, and ultrasound systems provide the bull's eye graphic, which displays segmental longitudinal strain: normal value is about $-20\%$, but the value of GLS increases in impaired systolic deformation, reaching positive values in LV aneurysm, where the segment shortening is absent [75].

Thanks to its features, GLS provides more accurate information. Gowsini et al. showed that each increase of 1% in infarct size measured by cardiac magnetic resonance corresponded to an impairment in GLS by 1.59% (95% CI 0.57–2.61, $p = 0.02$), while no significant association was found between infarct size and LVEF [76].

Moreover, GLS also demonstrated to be a better predictor of mortality than LVEF in aortic stenosis [77] and HF [78,79], especially with LVEF >35% [80]. In addition, in the general population, LV dysfunction assessed with GLS is four times more common than LV dysfunction assessed with LVEF [81], and it is an independent predictor of CV morbidity and mortality [81,82].

These features have made GLS a very useful technology in patients with cancer, who benefit most from an accurate cardiovascular evaluation before starting treatment to reduce incidence of CV events [83].

Significant GLS impairment is more frequent after cardiotoxic cancer therapy and occurs before reduction in other echocardiographic parameters, such as LVEF [84]. Moreover, it is an independent early stronger predictor of CTRCD in these patients [85].

GLS showed to be superior to LVEF in diagnosing subclinical cardiotoxicity in 100 breast cancer patients who completed anthracycline treatment without experiencing HF symptoms [86]. In addition, in myocarditis due to the use of immune checkpoint inhibitors, GLS was more strongly associated with major adverse cardiac events (MACE) compared to LVEF [24].

Fourati et al. [87] assessed in 103 patients with breast cancer a significant impairment of the LV GLS 3 months after they received radiation therapy, and this alteration persisted 6 months and 1 year post-radiation therapy. Interestingly, the mean dose of radiation was significantly higher in altered segments (6.7 ± 8.8 Gy–7.8 ± 8.9 Gy vs. 4.9 ± 7.9–5.4 ± 8.2 Gy; $p < 0.05$), suggesting that the radiation dose is correlated with subclinical alteration in LV segments, and segmental delineation may be a more effective alternative than global heart delineation in order to reduce the risk of CTRCD.

Moreover, Chen et al. [88] investigated alterations in echocardiographic parameters after radiation therapy in patients with locally advanced non-small cell lung cancer (NSCLC): no significant reductions were assessed in LVEF, LV end-diastolic diameter, LV end-diastolic volume and LV end-systolic volume, while a significant change was observed in GLS immediately after and 6 months after radiation therapy ($-18.1 \pm 2.5$ at baseline, $-15.7 \pm 2.6$ immediately after, $-16.2 \pm 2.3$ 6 months after radiation therapy, $p < 0.001$). In addition, the percentage change in GLS at 6 months showed to be an independent predictor for all-cause mortality (HR = 1.202, 95% CI: 1.095–1.320, $p < 0.001$), and a change in GLS $\geq 13.65\%$ showed in ROC analysis 65.9% sensitivity and 85.2% specificity for predicting mortality in patients with NSCLC [88].

A recent meta-analysis by Oikonomou et al. indicated a relative impairment of >15% in GLS as the threshold to predict future significant CTRCD [89].

Therefore, the last ESC guidelines on cardio-oncology have indicated this value as the cut-off for the decisional algorithm. Moreover, GLS is considered the preferred technique to detect and confirm cardiac dysfunction, and its use is recommended in all patients with cancer (class I C) [12].

Interestingly, GLS could guide the choice of patients who should start cardioprotective therapy before starting cardiotoxic therapy [90]: in a recent randomized controlled trial including 307 patients with cancer, Thavendiranathan et al. showed that starting cardioprotective therapy guided by GLS impairment led to a significantly lower reduction in LVEF at 1-year follow-up, compared to those patients who started cardioprotective therapy guided by LVEF reduction [91].

Furthermore, a 2016 study by Nolan et al. [92] compared the cost-effectiveness of three different cardio-protection strategies based, respectively, on diagnosis of LVEF-defined cardiotoxicity, universal cardio-protection for all patients and a strain-guided cardioprotection strategy, using an index case and scenario analyses with a Markov model. In the reference case of a 49 year-old woman with breast cancer treated with anthracyclines and trastuzumab, a GLS-guided cardio-protection strategy provided additional QALY at less cost compared with other strategies, also producing the highest value of 5-year survival [92].

Advantages and limitations of GLS are summarized in Table 1.

**Table 1.** Advantages and limitations of GLS. CTRCD: cancer-therapy-related cardiac dysfunction; GLS: global longitudinal strain; LV: left ventricle; ROI: region of interest.

| Advantages | Limitations |
| --- | --- |
| Cost-effective | Not available for all ultrasound devices |
| High reproducibility | Low image quality reduces measurements accuracy |
| Minimal angle dependence | Experienced operator required |
| Ability to measure strain in multiple LV segments from a single acquisition | Inter-vendor variability in strain measurements |
| High predictive power for mortality | Error in ROI estimation leads to GLS underestimation |
| Earlier detection of subclinical LV dysfunction | Increased value in response to preload increase |
| It is an independent stronger predictor of CTRCD | |

## 4. Effects of Exercise Training on Global Longitudinal Strain (GLS) in Patients with Cancer and Cardiac Disease

Because of its prognostic value [80,82] and its capacity to provide more accurate information, GLS has been used as end point in recent trials investigating ET effects in patients with CVD [93]. However, these trials have yielded conflicting results.

Favorable ET effects on strain parameters have been demonstrated in patients with arterial hypertension.

Sahin et al. [94] showed a significant improvement in GLS and a reduction in blood pressure in 30 hypertensive patients who underwent a cardiac rehabilitation program. Similar results were obtained by O'Driscoll et al. [95], who demonstrated significant improvement in GLS ($-2.3 \pm 2\%$; $p < 0.001$) and global work efficiency and a reduction in global wasted work in unmedicated hypertensive patients undergoing isometric exercise training.

In patients with recent ST-segment elevation myocardial infarction (STEMI) and LVEF > 45%, Malfatto et al. [96] demonstrated that cardiac rehabilitation could improve

GLS and other echocardiographic parameters, thus counteracting abnormalities of LV function presenting after acute myocardial infarction.

In addition, D'Andrea et al. [97] showed that an HIIT rehabilitation program favored a significant improvement in GLS compared to an MCT program ($-17.8 \pm 3.8$ vs. $-15.4 \pm 4.3$; $p < 0.01$) in patients with recent acute coronary syndrome (ACS). Furthermore, GLS resulted in a stronger association to functional capacity parameters, such as peakVO2 (R = $-0.40$; $p < 0.01$), compared to LVEF [97].

However, ET appears to allow patients with recent myocardial infarction to reach normal GLS value compared to healthy individuals, even using different training protocols [98].

A sub-study from SAINTEX-CAD (a large trial which applied aerobic interval training or aerobic continuous training to 200 patients with LVEF > 40 and recent coronary revascularization) showed a slight but nonsignificant GLS improvement in patients undergoing interval training compared to continuous training [99]. Moreover, higher peak VO$_2$ at baseline resulted in a non-significant correlation with GLS after multivariate stepwise regression [99]. Unfortunately, a comparison with a control group that did not undergo exercise training was not planned.

On the other hand, in patients with recent STEMI, Eser et al. [100] reported a worsening in GLS for those in the HIIT group at 1-year follow-up after completion of the training program. However, this finding remains unclear and deserves further investigation.

Since impairment in GLS has been demonstrated to occur early before signs of cardiotoxicity from anti-cancer treatments [84], any finding of improvement in GLS due to ET in patients with cancer could prove even more strongly its effectiveness in minimizing CTRCD.

So far, CORE showed to improve CRF and quality of life and reduce CV risk factors in patients living with cancer [52,53,65], but there is scarce evidence about its effects on cardiac remodeling.

Hojan et al. [101] showed that an endurance and strength exercise program performed 3–6 months after trastuzumab treatment was effective in counteracting the reduction in LVEF, which occurred in the control group. Conversely, no significative impairment in GLS was found in either the exercise or the control group.

Howden et al. [102] showed that exercise training before starting anthracycline therapy attenuated functional decline, but GLS did not change significantly in any participant both in the exercise and the usual care group. Therefore, the authors suggested that GLS could not be useful to identify increasing risk. However, in this study, the population selected did not present high CV risk at entry, and the patients self-selected to participate in exercise training or usual care.

Kirkham et al. [103] found positive effects of an exercise bout performed 24 h before every doxorubicin administration on cardiac output, resting heart rate and decreased vascular resistance, but no effects on GLS or LVEF. However, since even in this case there was no change in GLS in any patient undergoing chemotherapy, the investigators hypothesized that their patients were protected from deterioration in GLS by the low blood pressure values found [103].

Although the above trials failed to demonstrate significant improvement in GLS, it would be possible to hypothesize that GLS might be useful for selecting patients who would derive the most benefits from a CORE program rather than predict outcome in all patients undergoing cardiotoxic therapy. In fact, GLS has been shown to be effective in predicting LVEF worsening in patients undergoing cardioprotective therapy [91].

Further information will be provided from the ongoing ONCORE trial [104], a prospective, randomized controlled trial enrolling 340 women with breast cancer at early stages scheduled to receive anthracycline and/or anti-HER2 therapy, randomly assigned to participation in a CR program group or a usual care group. This study will use resting LVEF and GLS as primary outcomes, aiming to provide more information about the contribution of an exercise-based CR program in preventing cardiotoxicity.

## 5. Limitations

This review has several limitations. To date, studies investigating the effects of exercise-based cardio-oncology rehabilitation programs above all assessed the effects on CRF. Therefore, there is lack of data concerning benefits on echocardiographic parameters and particularly on GLS, which instead have been demonstrated in other CVDs.

Moreover, although GLS has been shown to be more sensitive in detecting the cardiotoxic effects of cancer treatments, there is still scarce evidence on its use to select which patients to refer for cardioprotective therapy, and there are no studies on its possible role in predicting which patients would benefit most from exercise-based cardiac rehabilitation programs.

In the socioeconomic field, there is a lack of studies assessing cost-effectiveness of cardiac rehabilitation in patients with cancer. Furthermore, echocardiography, and therefore also GLS, is an operator-dependent method; therefore, studies carried out in different centers could be affected by this limitation.

Another limitation concerns cardiac rehabilitation programs for cancer patients. In fact, despite recent growing interest and the publication of guidelines by the major scientific societies, exercise-based cardio-oncology rehabilitation programs are scarce, and there are no standardized protocols. Furthermore, these patients are particularly vulnerable. Fever and neutropenia are common side effects of cancer treatment, so their participation at hospital setting rehabilitation programs might expose them to a high risk of acquiring infections; in addition, they may develop anemia or other disorders which impair CRF, requiring flexibility in exercise programs tailored to their characteristics.

Finally, the patients examined can be very different from each other, and comparisons are difficult. Not only do they have different types of cancer, but the same cancer may have different characteristics in each patient, such as the expression of specific receptors (e.g., overexpression of HER2 or estrogen receptors in breast cancer). Each cancer therapy has a different cardiotoxic action, but in none of the examined studies was a specific weight assigned to the treatments used.

## 6. Conclusions

CVD has become the first cause of morbidity and mortality in patients living with cancer who have undergone cardiotoxic treatment. For this reason, a new interdisciplinary specialty was created: cardio-oncology includes experts dedicated to recognizing and treating CVD at its onset in patients with cancer.

GLS is a relatively inexpensive and accessible ultrasound technique, which provides additional information on the risk of developing cardiac dysfunction following cardiotoxic therapy and therefore may help to prevent CVD in patients living with cancer.

CORE programs have been shown to counteract most of the damage caused by cardiotoxic therapy, reducing CV risk factors and improving or otherwise mitigating impairment in CRF.

However, to date, there are still some aspects that need to be ameliorated: the absence of standardized protocols, the uncertainty about whether it is better to start ET before or after cardiotoxic therapy and which patients are the ideal candidates (actually, guidelines recommended cardiac rehabilitation only in patients at higher CV risk).

Moreover, positive effects of cardiac rehabilitation on CRF have been demonstrated, but efficacy data on cardiac remodeling is lacking.

In conclusion, it should be noted that most of the studies presented in the review refer to patients with coronary artery disease and the little evidence on GLS in patients with cancer undergoing CORE led to weak and contrasting results. Further studies will be useful to ascertain whether GLS can help select the patients who benefit most from CORE programs, as it has already been shown to do for medical therapy. Therefore, results from ongoing trials which are evaluating the effect of ET on GLS in patients with cancer are eagerly awaited.

**Author Contributions:** Conceptualization, G.C., F.P.I. and F.G.; data curation, A.C., M.P., E.T. and A.D.L.; methodology, F.P.I. and F.G.; validation, F.P.I., M.C. and F.G.; investigation, G.C.; resources, F.G., E.V., A.D. and C.T.; writing—original draft preparation, G.C., F.P.I. and F.G.; writing—review and editing, M.C., A.D., C.V. (Carlo Vigorito) and C.V. (Carmine Vecchione); visualization, G.C. and A.D.; supervision, C.V. (Carlo Vigorito) and F.G. All authors have read and agreed to the published version of the manuscript.

**Funding:** This research received no external funding.

**Institutional Review Board Statement:** Not applicable.

**Informed Consent Statement:** Not applicable.

**Data Availability Statement:** Data sharing not applicable.

**Conflicts of Interest:** The authors declare no conflict of interest.

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
