# Peer review of "Potential Role of Global Longitudinal Strain in Cardiac and Oncological Patients Undergoing Cardio-Oncology Rehabilitation (CORE)"

_clinpract, doi:10.3390/clinpract13020035_

Round 1
Reviewer 1 Report
The article discusses how certain cancer treatments can cause heart problems and how cardio-oncology rehabilitation (CORE) is a new interdisciplinary field aimed at improving the cardiovascular health of cancer survivors. The CORE involves correcting risk factors, drug therapies, and structured exercise programs. The article highlights the use of global longitudinal strain (GLS), an imaging modality, to evaluate the effectiveness of exercise training on cardiac function in cancer patients undergoing CORE programs. The review summarizes the evidence on the usefulness of GLS in this context.
The review is well-written; only minor corrections are required:
- I prefer using "patients with cancer" rather than "cancer patients" to avoid stigmatizing them and use "patients living with cancer" rather than "cancer survivors."
- Table 1: uses "cost-effective" instead of "not expensive."
- any data on the economic burden of CVD in patients with cancer or the cost-effectiveness of cardiac rehabilitation programs?
- please add "Limitations" section before the conclusion to discuss the limitations faced while conducting this review
Author Response
The article discusses how certain cancer treatments can cause heart problems and how cardio-oncology rehabilitation (CORE) is a new interdisciplinary field aimed at improving the cardiovascular health of cancer survivors. The CORE involves correcting risk factors, drug therapies, and structured exercise programs. The article highlights the use of global longitudinal strain (GLS), an imaging modality, to evaluate the effectiveness of exercise training on cardiac function in cancer patients undergoing CORE programs. The review summarizes the evidence on the usefulness of GLS in this context.
The review is well-written; only minor corrections are required:
- I prefer using "patients with cancer" rather than "cancer patients" to avoid stigmatizing them and use "patients living with cancer" rather than "cancer survivors."
We are thankful to Reviewer for Her/His suggestions. Accordingly, these terms have been changed in the revised manuscript
- Table 1: uses "cost-effective" instead of "not expensive."
We are thankful to Reviewer for Her/His suggestions. Accordingly, Table 1 has been modified.
- any data on the economic burden of CVD in patients with cancer or the cost-effectiveness of cardiac rehabilitation programs?
As requested, data on the economic burden of CVD in patients with cancer have been added to the revised manuscript. There is no information about cost effectiveness of cardiac rehabilitation programs in cancer patients, but we added data about cost effectiveness of cardiac rehabilitation and strain-guided cardioprotection strategy.
- please add "Limitations" section before the conclusion to discuss the limitations faced while conducting this review
According to Reviewer’ suggestions, a “Limitation” section has been added to the revised manuscript.
Reviewer 2 Report
In the first part of the article, chest radiation is cited as a possible reason for cardiotoxicity in these patients. however, I didn’t find any other data regarding chest radiation in the article. Although modern radiotherapy techniques have significantly lowered the frequency of cardiac toxicity (especially in breast patients, who often undergo cardiotoxic therapies such as anthracyclines or monoclonal antibodies before starting radiotherapy), data are still find in literature, linked to old radiation techniques. Since is more and more frequent in our daily practice to give radiation therapy to mediastinal mass, lung cancer or breast cancer patients with systemic or oral therapies which may have cardiotoxicity, I think would be more beneficial to take that in account in the article.
Author Response
In the first part of the article, chest radiation is cited as a possible reason for cardiotoxicity in these patients. however, I didn’t find any other data regarding chest radiation in the article. Although modern radiotherapy techniques have significantly lowered the frequency of cardiac toxicity (especially in breast patients, who often undergo cardiotoxic therapies such as anthracyclines or monoclonal antibodies before starting radiotherapy), data are still find in literature, linked to old radiation techniques. Since is more and more frequent in our daily practice to give radiation therapy to mediastinal mass, lung cancer or breast cancer patients with systemic or oral therapies which may have cardiotoxicity, I think would be more beneficial to take that in account in the article.
We are thankful to Reviewer for Her/His suggestions. As suggested, data about effects of radiation therapy on LV function and sensivity of GLS for these alterations have been added to the revised manuscript.